# *Mycobacterium tuberculosis* Affects Protein and Lipid Content of Circulating Exosomes in Infected Patients Depending on Tuberculosis Disease State

**DOI:** 10.3390/biomedicines10040783

**Published:** 2022-03-27

**Authors:** Fantahun Biadglegne, Johannes R. Schmidt, Kathrin M. Engel, Jörg Lehmann, Robert T. Lehmann, Anja Reinert, Brigitte König, Jürgen Schiller, Stefan Kalkhof, Ulrich Sack

**Affiliations:** 1College of Medicine and Health Sciences, Bahir Dar University, Bahir Dar P.O. Box 79, Ethiopia; 2Institute of Medical Microbiology and Epidemiology of Infectious Diseases, Faculty of Medicine, Leipzig University, 04103 Leipzig, Germany; brigitte.koenig@medizin.uni-leipzig.de; 3Institute of Clinical Immunology, Faculty of Medicine, Leipzig University, 04103 Leipzig, Germany; ulrich.sack@medizin.uni-leipzig.de; 4Department of Preclinical Development and Validation, Fraunhofer Institute for Cell Therapy and Immunology, 04103 Leipzig, Germany; johannes.schmidt@izi.fraunhofer.de (J.R.S.); joerg.lehmann@izi.fraunhofer.de (J.L.); stefan.kalkhof@hs-coburg.de (S.K.); 5Institute for Medical Physics and Biophysics, Faculty of Medicine, Leipzig University, 04107 Leipzig, Germany; kathrin.engel@medizin.uni-leipzig.de (K.M.E.); juergen.schiller@medizin.uni-leipzig.de (J.S.); 6Fraunhofer Cluster of Excellence Immune-Mediated Diseases CIMD, 04103 Leipzig, Germany; 7Department of Diagnostics, Fraunhofer Institute for Cell Therapy and Immunology, 04103 Leipzig, Germany; robert.lehmann@izi.fraunhofer.de; 8Institute of Anatomy, Histology and Embryology, Faculty of Veterinary Medicine, Leipzig University, 04103 Leipzig, Germany; anja.reinert@vetmed.uni-leipzig.de; 9Institute for Bioanalysis, University of Applied Science Coburg, 96450 Coburg, Germany

**Keywords:** exosomes, lipids, proteins, *Mycobacterium tuberculosis*, plasma, tuberculosis

## Abstract

Tuberculosis (TB), which is caused by the bacterium *Mycobacterium tuberculosis* (*Mtb*), is still one of the deadliest infectious diseases. Understanding how the host and pathogen interact in active TB will have a significant impact on global TB control efforts. Exosomes are increasingly recognized as a means of cell-to-cell contact and exchange of soluble mediators. In the case of TB, exosomes are released from the bacillus and infected cells. In the present study, a comprehensive lipidomics and proteomics analysis of size exclusion chromatography-isolated plasma-derived exosomes from patients with TB lymphadenitis (TBL) and treated as well as untreated pulmonary TB (PTB) was performed to elucidate the possibility to utilize exosomes in diagnostics and knowledge building. According to our findings, exosome-derived lipids and proteins originate from both the host and *Mtb* in the plasma of active TB patients. Exosomes from all patients are mostly composed of sphingomyelins (SM), phosphatidylcholines, phosphatidylinositols, free fatty acids, triacylglycerols (TAG), and cholesterylesters. Relative proportions of, e.g., SMs and TAGs, vary depending on the disease or treatment state and could be linked to *Mtb* pathogenesis and dormancy. We identified three proteins of *Mtb* origin: DNA-directed RNA polymerase subunit beta (RpoC), Diacyglycerol O-acyltransferase (Rv2285), and Formate hydrogenase (HycE), the latter of which was discovered to be differently expressed in TBL patients. Furthermore, we discovered that *Mtb* infection alters the host protein composition of circulating exosomes, significantly affecting a total of 37 proteins. All TB patients had low levels of apolipoproteins, as well as the antibacterial proteins cathelicidin, Scavenger Receptor Cysteine Rich Family Member (SSC5D), and Ficolin 3 (FCN3). When compared to healthy controls, the protein profiles of PTB and TBL were substantially linked, with 14 proteins being co-regulated. However, adhesion proteins (integrins, Intercellular adhesion molecule 2 (ICAM2), CD151, Proteoglycan 4 (PRG4)) were shown to be more prevalent in PTB patients, while immunoglobulins, Complement component 1r (C1R), and Glutamate receptor-interacting protein 1 (GRIP1) were found to be more abundant in TBL patients, respectively. This study could confirm findings from previous reports and uncover novel molecular profiles not previously in focus of TB research. However, we applied a minimally invasive sampling and analysis of circulating exosomes in TB patients. Based on the findings given here, future studies into host–pathogen interactions could pave the way for the development of new vaccines and therapies.

## 1. Introduction

According to a World Health Organization (WHO) report in 2021, tuberculosis (TB) was the biggest cause of death from a single infectious agent until the coronavirus (COVID-19) pandemic, having even surpassed HIV/AIDS [1]. The disease most commonly affects the lungs (pulmonary TB), but the pathogen can also affect other parts of the body, including the lymph nodes (TB lymphadenitis). *Mycobacterium tuberculosis (Mtb)* is the etiologic agent of TB, and its ability to penetrate immune cells and develop a niche by evading the host’s defense mechanisms is crucial to its pathogenic success [2,3]. The first step in the *Mtb* infection process is phagocytosis of *Mtb* by lung-resident alveolar macrophages [4]. *Mtb* can replicate in these cells, escape from the phagosome, and transmit the infection by migrating to neighboring lymph nodes [3]. The scope of the TB epidemic is so severe and damaging that ending TB by 2030 is one of the UN’s Sustainable Development Goals [5]. Despite numerous reports on *Mtb* strains, comprehensive information on host genetic and pathogenetic variables for strain proliferation, as well as host–pathogen preferences of the indicted and suspected *Mtb*, is lacking. Therefore, research breakthroughs (e.g., a new vaccine, accurate and reliable diagnostics) are needed to rapidly reduce TB incidence worldwide.

Exosomes, small unilamellar vesicles (30–150 nm in diameter) with a diverse cargo and composition (lipid, proteins, small RNAs) that are shed by cells, are increasingly recognized as a means of cell-to-cell contact and exchange of soluble mediators, an additional mechanism of intercellular communication [6], including the interaction between microorganisms and infected humans. Moreover, they are involved in the pathogenesis of infections by delivering infectious material to the recipient cells [7]. In the case of TB, exosomes are released by the bacillus and infected cells under healthy and pathological situations during *Mtb* infection [8,9,10]. These exosomes transport cytoplasmic and membrane-associated proteins, lipids, glycoconjugates, and nucleic acids [11]. Using biochemical assays and mass spectrometry (MS)-based methods, protein and lipid contents of exosomes from various sources have already been investigated and revealed exosome-specific patterns [12,13,14]. Lipids, a broad group of amphiphilic molecules, play important roles in cellular energy storage, structure, and signaling [15]. They are main components of all biological membranes, organelles and other structures derived thereof, such as extracellular vesicles (EVs) [16]. Lipidomics has been shown to be an effective method for studying disease processes and biomarkers in a variety of disorders [17]. For example, Tao et al. discovered that various phospholipids in plasma EVs were linked with pancreatic cancer at the tumor stage and overall survival using untargeted and targeted lipidomics [18]. An exosome proteomic study has revealed a number of proteins that are expressed in a pathogenesis of *Mtb* [8,19]. Exosomal proteins from patients with active TB are known to promote mycobacterial adhesions [20], and contribute to *Mtb* intracellular survival [21,22]. Those findings were also confirmed in a mouse model of *Mtb* infection [23]. Until now, however, studies of the lipid and protein content of circulating exosomes within the context of host–pathogen co-evolution in *Mtb* expansion in TB patients’ plasma are sparse.

Therefore, the aim of this study was to determine both host- and *Mtb*-derived exosome proteins and lipids from the plasma of TB patients and to assess treatment and disease state influence their abundance. Therefore, the protein and lipid compositions are compared between patients with TB lymphadenitis (TBL), i.e., an inflammation of lymph nodes due to an acute or chronic *Mtb* infection, and patients with pulmonary TB (PTB) before and after anti-TB treatment. The results give pathogenetic insights into the graft–host interaction and could eventually lead to improvements in vaccine development, diagnosis, and therapy response monitoring.

## 2. Materials and Methods

### 2.1. Preparation of Peripheral Blood Plasma

Subjects have been enrolled in the Debre Tabor Referral Hospital, Debre Tabor, Ethiopia. Peripheral blood samples have been acquired from each study subject (i.e., healthy controls, pulmonary TB, tuberculous lymphadenitis and pulmonary TB patients after anti-TB treatment, hereafter designated as HC, PTB, TBL, and Rx), respectively. Patients were diagnosed with TB in line with the Ministry of Health of Ethiopia country-wide guideline using the sputum Xpert MTB/RIF Assay test and histopathology of body fluids for PTB and TBL, respectively. Blood was collected and prepared in ethylenediaminetetraacetic acid, sodium salt (EDTA)-coated tubes. Nine milliliters of blood were collected, and samples were gently inverted five times and processed within 30 min of collection following two consecutive centrifugation steps at 2000× *g* for 20 min at room temperature (RT) and 10,000× *g* for 30 min at 4 °C. Each plasma sample was then transferred into a clean tube, aliquoted, and stored at −80 °C until use.

The study was conducted with the consent of the patients after obtaining an institutional ethical clearance from research and publication committee ethical review board of College of Medicine and Health Sciences, Bahir Dar University, Bahir Dar, Ethiopia (CMHS/IRB01-008) and the Faculty of Medicine, Leipzig University, Leipzig, Germany (250/18-ek). Sociodemographic characteristics of study participants are provided in Appendix A.

### 2.2. Exosome Isolation by Size-Exclusion Chromatography and Ultrafiltration

The 35 nm qEV original size exclusion chromatography (SEC) columns (Izon 35, Izon Science, Christchurch, New Zealand) were utilized according to the manufacturer’s instructions. Briefly, the columns were equilibrated with 1 × PBS before being loaded with 0.5 mL of the plasma sample. The 20% ethanol (EtOH) storage buffer was gradually removed. After that, the columns were washed three times with 5 mL of 1× PBS, and then the 0.5 mL plasma samples were placed onto the column, followed by 0.5 mL of 1× PBS five times in a stepwise fashion. We used exosome-enriched fractions 7 and 8 according to the manufacturer’s general usage guideline. PBS (15 mL) was used to flush the column. All fractions 7 and 8 were combined and concentrated (12,000× *g* for 10 min at RT) using Amicon Ultra Vivaspin 500 centrifugal concentrators (Merck KGaA, Darmstadt, Germany) before being stored at −80 °C until further use.

### 2.3. Nanoparticle Tracking Analysis

The nanoparticle tracking analysis (NTA), i.e., the determination of the size and concentration of plasma-derived isolated exosomes, was performed using a NanoSight LM10 (Malvern Panalytical Ltd., Malvern, UK) equipped with a 450 nm laser, carried out according to the manufacturer’s protocols. The samples were diluted 1:200 in sterile 0.9% NaCl. Each experiment was repeated three times (each for a total capture time of 60 s with a frame rate of 25 frames/s and a camera level of 11). The cumulative percentage of nanoparticles was calculated using the nanoparticle size distribution curve, refractive index, and relative nanoparticle concentration of a certain size for each sample. The data was processed using NTA 3.0 software with a detection threshold of 4 at a temperature control of 25 °C. 

### 2.4. Transmission Electron Microscopy

Transmission electron microscopy (TEM) was used to visualize exosomes, as previously described by Zhupanyn and colleagues [24]. Briefly, each exosome suspension was fixed with 2% formaldehyde and 2% glutaraldehyde in 0.1 M PBS (pH 7.4) for 20 h at 4 °C. Exosomes were adhered to formvar-coated 300-mesh nickel grids by incubation for 10 min in a droplet of suspension (concentration of 2.5 × 10^6^ exosomes/µL). Grids were washed three times in dH_2_O for 1 min and excess liquid was removed by blotting with filter paper. The samples were negatively stained with 1% aqueous uranyl acetate for 1 min. The grids were again blotted on filter paper to remove excess stain, air-dried, and stored in an exicator. Exosomes were visualized by TEM by using a Zeiss Libra 120 transmission electron microscope (Zeiss, Oberkochen, Germany) in the bright field mode at 80 keV. Images were taken with a 4-MP CCD camera equipped with an Yttrium Aluminum Garnet (YAG) scintillator and ISP software (TRS Tröndle, Moorenweis, Germany).

### 2.5. Flow Cytometry

The MACSPlex exosome kit, human (Miltenyi Biotec B.V. & Co. KG, Bergisch Gladbach, Germany) was used to perform flow cytometric exosome analysis according to the manufacturer’s overnight incubation protocol. Exosomes derived from plasma were diluted in MACSPlex buffer to a final volume of 120 µL, mixed with 15 µL of exosome capture beads coated with 37 different antibodies, and placed in one well of a microplate, where they were incubated at 450 rpm overnight at RT in the dark on an orbital shaker. After washing, the mixture was incubated in an orbital shaker for 1 h at RT with 15 µL of a master mix of anti-CD9, anti-CD63, and anti-CD81 antibodies, recognizing the respective cell surface glycoproteins of the tetraspanin family, CD9 and CD81, which bind to antigen-presenting cells (APC). A BD FACSCantoTM II flow cytometer (BD Biosciences, Heidelberg, Germany) was used for the analysis. The median APC signal intensity of each unique population of single beads was standardized to the average of the anti-CD9, anti-CD63, and anti-CD81 beads after the values of the control (buffer only) were subtracted from the investigated samples. Surface marker values that were less than the measurement threshold for the appropriate control antibody included in the kit were regarded as negative.

### 2.6. Lipid Extraction

Lipid extraction of exosomes derived from 9HC, 13 PTB, 13 TBL patients, and 6 Rx-patients was performed according to Bligh and Dyer [25] with slight modifications, as described in the following. Exosome preparations were mixed with 1 mL of methanol and 1 mL of chloroform in glass vials, and then incubated at 750 rpm for 30 min on an orbital shaker at room temperature (RT). One ml of water was added, and the samples were again incubated at 750 rpm for 30 min at RT. For phase separation, samples were centrifuged at 2500 rpm for 7 min at RT. The lower organic phases were carefully withdrawn with a glass syringe and lipid extraction was repeated with another 1 mL of chloroform, plus vortexing and centrifugation as described above. The two organic phases of one sample were combined and the solvent was removed by evaporation in a Jouan centrifugal evaporator 1022 (Thermo Scientific, Waltham, MA, USA). Lipids were dissolved in 40 µL chloroform and transferred into a 1.5-mL vial with a glass insert. The samples were stored at −20 °C until further analysis.

### 2.7. Lipid Analysis by High-Performance Thin-Layer Chromatography and Electrospray Ionization

Lipid extracts (20 µL) were automatically spotted onto an HPTLC glass back plate (Merck KGaA, Darmstadt, Germany) by using a Linomat (CAMAG, Muttenz, Switzerland) and separated with chloroform/ethanol/water/triethylamine (30:35:7:35, by vol.) as the mobile phase. After air-drying for 15 min, the HPTLC plate was dipped into primuline solution (Direct Yellow 59, Sigma-Aldrich, Taufkirchen, Germany, 50 mg/l in acetone/water (80:20, by vol.)). Lipids were visualized under UV light (366 nm) and spots were marked with a pencil. The lipids in each spot were automatically eluted by a Plate Express™ TLC plate reader (Advion, Ithaca, NY, USA) with methanol as solvent and directly analyzed by electrospray ionization mass spectrometry (ESI MS) on an Amazon SL mass spectrometer (Bruker Daltonics GmbH, Bremen, Germany) with the following settings: Spray voltage 4.5 kV, end plate offset 500 V, nebulizer gas 7 psi, drying gas (N_2_) 3 L/min, capillary temperature 200 °C. For data acquisition and subsequent analysis of lipid spectra, the software “Trap Control” and “Data Analysis” version 4.1 (Bruker Daltonics GmbH) were used, respectively.

### 2.8. Sample Preparation for Proteomics and Raw Data Acquisition

The protein content was extracted from plasma-derived exosomes of six PTB patients, three TBL patients, three treated PTB patients, and six HC. Combined exosome-containing SEC fractions were concentrated and lysed on a centrifugal filter unit (Vivacon 500, 10K MWCO, Sartorius, Göttingen, Germany) using a modified RIPA buffer (25 mM Tris-HCl, pH 7.4; 125 mM NaCl; 1% Triton X-100 (*v*/*v*); 1% Na-Deoxycholate (*w*/*v*); 0.1% SDS (*w/v*) in ddH_2_O) by incubation at 4 °C and additional sonication treatment. Lysates were centrifuged at 14,000× *g* for 5 min to clear the fraction. The supernatant was subsequently used as the protein fraction. Protein concentration was determined using the Pierce BCA Protein Assay (Thermo Fisher Scientific, Rockford, IL, USA). Extracted proteins were processed by filter-aided sample preparation as previously published [26]. Briefly, proteins were washed in a urea buffer (8 M urea in 100 mM triethylammonium bicarbonate, pH 8.5) and subsequently reduced and alkylated in 10 mM tris (2-carboxyethyl)phosphine (TCEP) and 55 mM iodoacetamide (IAA), respectively. Proteolytic cleavage to peptides was carried out by incubating the proteins with a Trypsin/Lys-C protease mix (Promega GmbH, Walldorf, Germany) overnight at 37 °C, at a molar protein/enzyme ratio of 50:1. Proteolytic peptides were recovered by ultrafiltration. These peptides were desalted using Pierce Peptide Desalting Spin columns (Thermo Fisher Scientific) according to manufacturer’s instructions followed by resuspension in 100 mM TEAB. Peptides were chemically labeled with 10-plex tandem mass tag kit (TMT, Thermo Fisher Scientific) according to manufacturer’s instructions and combined into two TMT batches, including a common reference consisting of equal amounts of all samples (Table 1). Finally, TMT batches were again desalted using Pierce Peptide Desalting Spin columns.

For sample acquisition, 1 µg of labeled peptides were injected into an Easy-nLC 1200 coupled to a Q Exactive HF mass spectrometer (both Thermo Fisher Scientific). Peptides were separated on a 20-cm analytical HPLC column (75 µm ID Pico Tip fused silica emitter, New Objective, Littleton, MA, USA) packed in-house using ReproSil-Pur C18-AQ 1.9-μm silica beads (Dr. Maisch GmbH, Ammerbuch, Germany), applying a 120-min multistep gradient from 10% solvent B to 90% solvent B at a constant flow rate of 200 nL/min. The mobile phases were prepared as 0.1% formic acid in water (solvent A) and 80% ACN, supplemented with 0.1% formic acid in water (solvent B). Eluting peptides were ionized by electrospray ionization at 3.2 kV. Data was acquired in data-dependent mode and controlled by XCalibur software (version 2.9). Survey scans were acquired with an orbitrap mass analyzer in a scan range of 300–1650 *m/z*, a mass resolution of 60,000, AGC target of 3 × 10^6^ and a maximum injection time of 25 ms. The top 10 most abundant precursor ions were selected for isolation (window of 0.7 *m/z*) and fragmentation by higher-energy collisional dissociation (normalized collision energy: 34). For MS2 scans in orbitrap mass analyzer, AGC target and maximum injection time were set to 1 × 10^5^ and 110 ms, respectively. A dynamic exclusion for MS2 scans was set to 30 s.

### 2.9. Proteomics Data Analysis and Database Integration

Proteomics raw data was analyzed using the software MaxQuant (version 1.6.3.3) and the integrated protein identification algorithm Andromeda [27]. Experimental spectra were matched against reference proteomes of *H. sapiens* (UP000005640, version 2021_04, 79,038 protein entries) and *M. tuberculosis* (strain ATCC 25618 / H37Rv, UP000001584, version 2021_04 3,993 protein entries). Trypsin was defined as a position-specific protease in fixed mode with a tolerance of up to two missed cleavages. Carbamidomethylation (cysteine) was set as fixed modification, whereas oxidation (methionine) and acetylation (protein N-terminus) were set as optional modifications. A mass tolerance of 20 ppm (first search and fragment ions) and 4.5 ppm (main search) was allowed. One unique peptide was required for protein inference controlling the false discovery rate (FDR) to 0.05 for peptide spectrum matches, peptides, and proteins. The proteomics MS data have been deposited to the ProteomeXchange Consortium via the PRIDE [28] partner repository with the dataset identifier PXD030883 and 10.6019/PXD030883. Exported proteins were filtered to exclude “potential contaminants”, “reverse”, and “only identified by site” entries. The top 100 most frequently identified vesicular proteins were retrieved from Vesiclepedia [29] and mapped to the proteins identified and filtered here in the present study. Gene ontology (GO) annotations (DIRECT terms) for all identified and filtered proteins were retrieved by DAVID Bioinformatic Resources [30]. Therefore, gene names of protein groups were deconvoluted and searched separately. Subsequently, unique mapped annotations were re-grouped to initial protein groups again. Term overrepresentation analyses and clustering were conducted. Overrepresented terms were filtered by EASE (a modified conservative Fisher’s exact test score) <0.01. Ten representative terms per category were selected by most included proteins and covering all clusters. Full analysis outputs are provided as Appendix A. Data normalization for quantitative proteomics analysis was carried out in the R environment (version 4.0.2). First, summed intensities of reporter ions per sample were normalized to correct systematic errors from sample loadings. Next, inter-batch correction per protein was carried out by normalizing the intensity to common reference channel, followed by trimmed mean of M values normalization (TMM), applying “calcNormFactors” function of “edgeR” package (version 3.32.1) [31].

### 2.10. Data Depiction and Statistical Analysis

Graphs showing exosome concentration and size distribution and the respective statistical analyses (one-way analysis of variance (ANOVA)) were performed using Microsoft Excel 2016 software. Descriptive data analysis was done to visualize differences within data sets. Differences with *p* ≤ 0.05 were considered to indicate statistically significant differences. In lipidomics analyses, the graphs and statistical analyses for the lipid composition of exosomes were designed and performed using GraphPad (version 8.4.3, GraphPad Software, San Diego, CA, USA). Data of lipid analyses were visualized as bar graphs showing the mean and the positive standard deviation or as dot-plots representing all single values. Statistical significance was determined using the unpaired t test and the Holm–Sidak method to correct for multiple comparisons with alpha = 0.05. Graphs and statistical analyses of proteomics data were designed and performed in R environment (version 4.0.2). For heatmaps, normalized intensities were scaled row-wise. The “ComplexHeatmap” package (version 2.4.3) was used. Further figures were prepared using “ggplot2” (version 3.3.5) and “ggpubr” (version 0.4.0) packages. Global statistically significant proteins were identified by applying ANOVA across all sample groups. Only proteins quantified in all samples were included. Resulting *p* values were Benjamini–Hochberg adjusted and proteins possessing an adjusted *p* < 0.01 were assigned as a differential abundant protein (DAP). Pairwise statistical analysis of patient groups was performed using the “DEqMS” package (version 1.6.0). Only proteins quantified in at least 3 samples of corresponding groups were included. Similar to ANOVA analysis, proteins possessing an adjusted *p* < 0.01 were assigned as DAP. Due to the low number of identified proteins and low intensities, replicates 1 and 4 of HC had to be excluded from statistical analyses. Functional network analysis of DAP, including network enrichment, was performed by STRING (version 11.5) [31].

## 3. Results

### 3.1. Exosomes from Patients’ Plasma Specimens Are Successfully Isolated by SEC

Exosomes were extracted by SEC and analyzed by NTA. The concentrations were found to be at 6.70 ± 0.44 × 10^8^, 2.92 ± 0.87 × 10^8^, 4.15 ± 1.41 × 10^8^, and 6.86 ± 1.94 × 10^8^ particles/mL for HC, PTB, TBL, and Rx, respectively (Appendix A). The mean size distributions of nanovesicles isolated from HC, PTB, TBL, and Rx were at 165.8 ± 6.9 nm, 175.8 ± 7.1 nm, 150.3 ± 6.8 nm, and 133.3 ± 20 nm, respectively (Appendix A). The mode size distribution of isolated exosomes was at 122.7 ± 5.4 nm for HC, 126.7 ± 5.8 nm for PTB, 124.2 ± 10.8 nm for TBL, and 99.6 ± 14.3 nm for Rx. In addition, the mean particle distribution for nanovesicles isolated from plasma indicated that 50% of the population lies below 159.3 ± 4.0 nm for PTB, 148.4 ± 3.4 nm for HC, 133.9 ± 5.8 nm for TBL, and 118.1 ± 17.3 nm for Rx. In general, the size distribution of vesicles showed a maximum size range (peaked at 126 nm, 132 nm, 122 nm, and 119 nm for HC, PTB, TBL, and Rx, respectively (Appendix A–F), which is in the typical range characteristic of exosomes.

### 3.2. TEM and Surface Marker Analysis Confirm Exosome Identity

TEM was used to determine size and morphology of exosomes isolated by SEC from human blood plasma. Our observations revealed that both preparations contained exosomes from 30 to 150 nm in diameter, confirming our findings from NTA. A representative image of the three similar experiments from each study group showing intact vesicles with classic exosomes morphology is presented in Figure 1. Human blood plasma-derived exosomes were investigated for exosome surface marker signatures by flow cytometry and confirmed our findings from NTA and TEM. This multiplex analysis of 37 frequently observed exosomal surface epitopes revealed strong signals for the exosome-associated markers CD9, CD63, and CD81 (Figure 2 and Appendix A).

### 3.3. Lipidomics Reveals Disease and Treatment State-Specific Molecular Features of Exosomes

The separation of the lipid fractions by HPTLC (Appendix A) revealed that the exosomes of all blood plasmas are mainly comprised of sphingomyelins (SM), phosphatidylcholines (PC), triacylglycerols (TAG), and cholesterylesters (CE). There were phosphatidylinositols (PI) and free fatty acids (FFA) detectable in all samples but at lower abundances. The amounts of lysophosphatidylcholines (LPC), phosphatidylserines (PS), and phosphatidylethanolamine (PE) in all samples were (if at all detectable) not sufficiently high to obtain interpretable ESI mass spectra. A densitometric evaluation of the TLC plates revealed significant differences in the ratios of SM/TAG and PC/TAG between the patient groups and HC (Appendix A). Another TLC separation for nonpolar lipids revealed a tendency towards higher CE/TAG ratios in TBL patients (*n* = 4; mean ratio: 1.69 ± 0.22) compared to HC (*n* = 4; 1.36 ± 0.18). The mean CE/TAG ratio in Rx patients (*n* = 3; 1.37 ± 0.02) was nearly equal to HC.

Detailed analysis of the PC, SM, PI, and FFA and TAG/CE spots by ESI MS showed differences in the fatty acyl composition of lipids derived from the different patient groups. For SM (Figure 3), significant changes between the different cohorts were obvious for the amount of C atoms representing the chain length of the bound fatty acyl moiety and/or the respective sphingoid base of the SM species (Figure 3A), as well as for the amount of double bonds (db) representing the degree of unsaturation of the fatty acyl moiety (Figure 3B). Even though SM34:1;O2 was the most abundant SM in all samples, with a mean ratio of approximately 30 to 40%, in general SM from TBL patients were longer with a higher degree of unsaturation, represented by a higher amount of the larger SM molecules, less SM with one, and more SM with two or three db. Treatment of PTB patients (Rx samples) affected the relative amounts of single SM, such as SM 32:1;O2, SM 34:2;O2, SM 36:2;O2, SM 40:2;O2, SM 39:2;O2, and SM 42:1;O2 (Figure 3C). The SM composition of HC differed from PTB and TBL patients, but there were no differences between HC and Rx.

The most pronounced differences between HC and patients could be detected in the PC moieties of the respective groups (Figure 4). There were no differences between the two patient groups as well as between HC and Rx, and PC 34:2 was the most abundant PC species in all samples (about 20–25%). The treatment of PTB patients influenced the overall PC chain lengths as well as the PC db content. Plasma-derived exosomes from Rx (and HC) samples showed longer chained PC molecules with more db compared to PTB and TBL patients (Figure 4A,B). This is also obvious for the amounts of single PC species, e.g., PC 32:0 and PC 34:1, which are lower, and PC 36:4, PC 38:6 and PC 38:5, which are in higher abundance after PTB treatment (Figure 4C). The amounts of the ether lipids PC o-40:2 and PC o-40:3 were lower after treatment compared to PTB and TBL patients.

The TAG fractions of all samples contained about 18 to 20% TAG 52:3 as the main TAG. TAGs in HC were shorter compared to patient samples, represented by higher amounts of TAG 48:1, TAG 48:2 and TAG 50:2, and lower amounts of TAGs with 54 C atoms. PTB and TBL patients differed significantly in regards to the concentration of TAG with 50 and 54 C atoms in their fatty acyl chains (Figure 5A). The latter was due to higher amounts of TAG 54:3, TAG 54:2 and TAG 54:1 (Figure 5C). The treatment of PTB patients caused an increase in the double bond content of TAG molecules (Figure 5B).

Interestingly, PI was found to be present in considerable amounts in all samples. There was no difference in the PI composition between PTB and TBL, but the treatment of PTB patients influenced the PI composition of plasma exosomes, leading to longer and higher unsaturated PI species (Appendix A), even beyond the respective amounts detected in HC. This difference was due to reduced concentrations of PI 34:2, PI 34:1, PI 36:1, and PI 36:2, as well as increased concentrations of PI 38:3 (Appendix A). PI 38:4 and PI 38:3 showed a tendency of being more abundant in Rx compared to HC, PTB, and TBL patients. Regarding CE, the relative concentration of CE 18:2 was higher after treatment compared to HC and PTB patients (Appendix A). CE 20:4 was significantly less abundant in TBL compared to PTB patients and HC. For FFA, there were no differences detected between the groups regarding the relative amounts (Appendix A).

### 3.4. Proteomics Identifies Host and Bacterial Proteins in Circulating Exosomes

To investigate whether *Mtb* infections affect the protein content of circulating exosomes, a TMT-based quantitative proteomics analysis was applied. In summary, 258 proteins were identified across all samples, of which 244 met the criteria for reliable quantification (Appendix A). The quantitative range spans nearly five orders of magnitude (Figure 6A). To estimate the purity of the collected vesicular fraction, we matched the proteins identified here with a list of the 100 most commonly observed vesicular proteins from the Vesiclepedia database. Thirty of these were also identified in this study, including the previously described exosomal marker proteins CD9 and LGALS3BP. However, 91 of the proteins identified here were also annotated as “secreted to blood” according to the Human Protein Atlas (Appendix A), possibly indicating sources of contamination from soluble plasma proteins. Gene ontology term overrepresentation and clustering analyses were performed (Appendix A). According to the gene ontology annotation of cellular component classification, 186 and 87 of the identified proteins were classified as extracellular exosome components and components of blood microparticles, respectively (Figure 6B). Other vesicular structures, e.g., high-density lipoprotein (HDL) particles and endosomes, were found to be enriched with 14 and twelve proteins, respectively. Further extracellular matrix and plasma membrane surface components as well as proteins in focal adhesions and cell–cell adherence junction sites, were enriched. Among the biological processes implicated (Figure 6C), proteins involved in classical complement activation and innate immunity are most frequently identified (44 and 41 proteins, respectively). Less frequently covered functions include antibacterial immunity by humoral response and phagocytosis. Furthermore, proteins involved in cell–cell adhesion and lipid transport were identified. Forty-two proteins share their molecular functions as serine-type endopeptidases (Figure 6D). Further, immunoglobulins (IG) were frequently identified, binding to antigens and IG receptors. Less frequently, proteins binding to heparin and integrin were identified. Thus, proteomics analysis further confirms the enrichment of exosomes by the here described SEC and ultrafiltration-based isolation. However, co-isolation of other vesicular blood particles is suggested from the data, being involved in complement activation and lipid transportation.

Remarkably, three proteins of *Mtb* origin, namely DNA-directed RNA polymerase subunit beta (RpoC), diacyglycerol O-acyltransferase (Rv2285), and formate hydrogenase (HycE) were also identified. While RpoC was below the quantification limit, Rv2285 and HycE were reliably quantified (Appendix A). Interestingly, HycE was identified as significantly more abundant in TBL patients in pairwise comparison to PTB patients. However, both Rv2285 and HycE were quantified in all patient groups, including healthy control patients not diagnosed with active tuberculosis. Thus, results regarding identification of *Mtb*-derived proteins have to be discussed and interpreted carefully in later studies.

### 3.5. M. tuberculosis Infection Alters Host Protein Composition of Circulating Exosomes

To investigate the host response upon *Mtb* infection and treatment, differentially abundant proteins (DAPs) in isolated exosomes were identified by ANOVA across all patient groups. Thirty-two DAPs were identified to possess disease state- or treatment-specific abundances (Figure 7A). Hierarchical clustering of proteins revealed three main branches, each characterized by specifically highly-abundant proteins at a certain disease state. Interestingly, samples of treated patients (Rx) do not possess specifically affected protein populations, but rather share features of both diseased but also untreated patient groups.

The first main branch comprises proteins that are more abundant in HC and depleted in *Mtb*-infected patients regardless of the disease or treatment state. This includes the antibacterial proteins Cationic Antimicrobial Protein (CAMP), Ficolin 3 (FCN3), and Scavenger Receptor Cysteine Rich Family Member (SSC5D), as well as eight apolipoproteins. The latter are forming a significant functional network (Figure 7B). The second main branch comprises proteins that are more abundant in PTB-derived samples, including integrins (ITGB1, ITGB3), the cell adhesion factor Vitronectin (VTN), the extracellular chaperone Clusterin (CLU), Alpha-1-antitrypsin (SERPINA1), and Haptoglobin (HP), which are likewise significantly connected by functional relationships (Figure 7C). The third main branch comprises proteins that are more abundant in TBL-derived samples. This includes six immunoglobulins (IGs), a complement component (C1R), the Spermatogenesis-Associated Protein 17 (SPATA17), and the Glutamate receptor-interacting protein 1 (GRIP1).

We found that protein profiles of exosomes from PTB and TBL patients were highly correlated, and 14 proteins were co-regulated at both disease states compared to healthy controls (Figure 7D and S6). However, eight and seven proteins were specifically affected in PTB or TBL, respectively. Whilst HP, Proteoglycan 4 (PRG4), Stomatin (STOM), CD151, Intercellular adhesion molecule 2 (ICAM2), Alpha-1-acid glycoprotein 1 (ORM1), Serum amyloid A-1 protein (SAA1), and Solute carrier family 2A3 (SLC2A3) are more abundant in PTB patients, several IG chain proteins and uncharacterized protein C9orf50 were enriched in TBL patients, indicating disease state-specific protein profiles of circulating exosomes.

To further address the effect of anti-TB treatment on host protein composition, a statistical pairwise comparison of treated and untreated PTB patients was included (Figure 7E). Eight proteins were identified as treatment-affected DAPs. Whilst GRIP1, the Ig chain IGHV4-28 and Myosin-9 (MYH9) were increased in abundance after anti-TB treatment, FCN3, HP, Serum amyloid A-4 protein (SAA4), Transmembrane protein 215 (TMEM215) and Apolipoprotein B-100 (APOB) were depleted. In summary, 37 proteins were identified as disease state- and/or treatment-dependent exosomal components upon *Mtb* infection, combining findings from ANOVA and pair-wise comparisons (Table 2).

## 4. Discussion

Our study characterizes the size, concentration, and surface marker signatures of exosomes isolated from the plasma of TB patients and extends the knowledge about the lipid and protein composition of human blood plasma-derived exosomes. In the present study, exosomes isolated from the plasma of healthy individuals and active TB patients (TBL, treated and untreated PTB) were demonstrated to have a size range of 30 to 150 nm [10,32,33]. The characteristic proteins CD9, CD63, and CD81 were all expressed on exosomes from active TB patients [9,34,35]. In addition, proteomics characterization confirmed the identity of exosomes from patient-derived plasma by identifying exosomal markers CD9, CD151, and LGALS3BP [36]. In summary, NTA, TEM, flow cytometry, and proteomics surface marker analysis confirmed the presence of an exosomal population, as all findings correspond to characteristics reported elsewhere. However, the co-isolation of other blood-derived particles, e.g., lipoproteins, immune complexes, or proteins of the complement system from patients’ plasma can be anticipated. This is in line with other reports of plasma-derived exosomes, regardless of the applied method of exosome isolation [37,38]. To the best of our knowledge, no gold-standard method that combines absolute purity of exosome extractions and feasibility in clinical settings has been previously established.

An inspection of the lipid contents of exosomes derived from TB patients (before and after anti-TB treatment) and healthy controls through lipidomics-based investigation revealed the presence of SM, PC, PI, FFA, TAG, and CE lipid fractions. This is in accordance with a previous report [10]. In these vesicles and in the lipid fractions of exosomes derived from TB patients’ plasma, the main SM and PC molecules are SM 34:1,O2 and PC 34:2. In a study reported 15 years ago, Subramanian et al. applied nuclear magnetic resonance (NMR) on lipid extracts from human intracranial tuberculomas and healthy brain tissue, and could show an altered lipid composition in extrapulmonary manifestation of TB [39].

In the present study, SMs from TBL had longer chain lengths and a higher number of double bonds compared to PTB. SMs interact with cholesterol and modulate membrane properties. They act as adhesion sites for proteins and play important roles in both signal transmission and cell identification [40]. SMs have already been involved in pathogenesis in TB and other infectious diseases [3]. *Mtb* influence host defense through modifying host membrane SMs and their metabolites, allowing them to survive and replicate [41].

Our data suggests significant amounts of TAG and CE in exosomes of infected patients, which is consistent with a study by Holert and colleagues [42]. They showed that *Mtb*, when grown on cholesterol- and fatty acid-containing medium, produced, and accumulated CE with the respective fatty acyl residue. Furthermore, we were unable to detect any odd-chained CEs. As blood plasma contains considerable amounts of cholesterol and free fatty acids, *Mtb* produces CE according to the host’s fatty acyl composition. Another study on lung biopsies of TB patients showed higher CE, TAG, and cholesterol levels after TLC separation of extracted lipids from caseous granulomas compared to healthy tissue [43]. This confirms our results that CE was hardly detectable in HC. Han et al. suggested blocking CE synthesis as a therapeutic target for anti-TB medications [44]. In contrast to the aforementioned study, we could not detect significant amounts of CE 20:3 in the exosomal preparations, which Han et al. suggested as an early marker for TB infection.

It has been shown that TAG levels were greater in infected patients but did not differ significantly from non-infected patients [45]. Our study shows differences between the different disease states (PTB vs. TBL) and changes in the TAG profile after treatment. Our results are supported by a recent publication from Shivakoti et al. [46], who reported on lipids associated with PTB treatment failure, and used patients with successful treatment as controls. According to their results, effective treatment leads to a reduction of, e.g., TAG 54:2, TAG 54:1, TAG 52:1, and PC 32:0, and to an increase in CE 20:4. In contrast to our results, they report decreases in SM 36:2 and CE 18:2. Discrepancies for the amount of certain lipids might stem from the different extraction methods used.

Overall, the presented data shows that PTB treatment leads to the restoration of the lipid moieties present before *Mtb* infection. It is possible that *Mtb* uses TAGs as energy storage during its long-term dormancy. As a result, TAG production could be a good target for new antilatency medications that prevent the organism from surviving dormancy and hence aid in TB control [47]. The pathogen collected from sputum had lipophilic inclusion bodies containing TAGs, which might be retained during dormancy or synthesized in the growing granuloma from fatty acids released from the deteriorating host tissue [48]. Experiments with putative TAG synthase disruptors are currently underway to determine whether TAG synthesis induction is essential for dormancy and reactivation. If this is the case, the TAG implicated in the process could be used to develop innovative medications to avoid dormancy and hence aid in the recovery process [47].

We could not detect phosphatidylserine (PS) in *Mtb*-derived exosomes, which is consistent with prior findings [46,49,50], implying that PS levels in exosomes from healthy plasma and TB patients are likely too low to be identified by our approach. However, this is in contrast to a study published in 2019 [51], which describes an enrichment of PS in extracellular vesicles of *Mtb* infected J774A.1 cells (mouse macrophages). Even though lipid extraction was also performed using chloroform/methanol, there seems to be a discrepancy between the settings. It should be noted that according to the authors and their TLC-based approach, J774A.1 cells exclusively present high amounts of PS and PE. This is in contrast to the known diversity of lipids in vertebrates, typically with only low amounts of PS and PE.

In summary, lipidomic investigations confirmed previous reports on *Mtb* effects on lipid profiles in patients that have already been linked to immune evasion, cell identification, energy storage, and dormancy and have been discussed as therapeutic targets. Of note, whilst previous studies are mainly based on biopsies, postmortem tissue samples, sputum, or in vitro experiments, we applied a minimally-invasive sampling of blood and exosome analysis, using no more than 500 µL patients’ plasma. Furthermore, we uncovered disease state- and treatment-specific lipid profiles.

In addition to differential lipid content profiles, quantitative proteomics revealed 37 proteins that are affected in different disease states and treatment conditions. Strikingly, the abundances of six different apolipoproteins were reduced after *Mtb* infection in all diseased patient groups. Apolipoproteins were previously discussed as TB biomarkers in different studies [52,53]. Jarsberg et al. found apolipoproteins associated with treatment response in PTB, but found differential expression based on geographical regions and socioeconomical parameters [54]. Thus, a link between TB susceptibility and nutrition status should be considered.

We further demonstrate that the antibacterial proteins CAMP (also known as cathelicidin), FCN3, and SSC5D are downregulated as a general feature in TB patients. This appears contradictive as those proteins are also known to bind bacterial pathogens in a defense response. Nevertheless, these findings are in line with previous reports. Rode et al. previously demonstrated an *Mtb*-related downregulation of cathelicidin in dendritic cells [55]. SSC5D was found to be downregulated in large transcriptomics analyses in active TB patients [56]. Gene polymorphism association studies found a putative relation between *FCN3* gene variants and clinical symptoms [57]. However, no direct evidence for *Mtb* affected regulation of FCN3 has been previously described. *Mtb* evolved different mechanisms to evade host immunity, therefore the regulation of those antibacterial proteins might be an additional layer.

This study additionally revealed disease state-specific protein profiles of circulating exosomes. Adhesion proteins such as integrins and ICAM2 were enriched in PTB patients compared to TBL patients. Macrophages use integrins to adhere to fibronectins and reach infection sites, and were found to be affected by *Mtb*-secreted proteins [58]. An association between the downregulation of integrins and a reduction in adhesion and migration was also found for dendritic cells in *Mtb* infection [59,60]. Similarly, *Mtb*-secreted proteins were found to bind to ICAM1 in inhibit immune cell invasion [61]. However, no such evidence was yet described for ICAM2. As those proteins are directly involved in defense response and downregulated in TBL patients, it might be considered that this host adaption is needed for *Mtb* to reach infection sites beyond lung tissue. In contrast, many IGs, as well as the complement factor C1R, were enriched in TBL patients compared to PTB patients in this study. Both have previously been associated with active TB infections [62]. Significantly increased levels of IGs have previously been described for extrapulmonary TB [63,64,65] and were utilized for the diagnostics of active TB [66,67].

Anti-TB treatment had only minor effects on exosome protein contents, with eight proteins found to be affected, including IGs (upregulated compared to untreated PTB), FCN3, HP, APOB, and SAA4 (all downregulated). Interestingly, and in contrast to our findings in lipidomic investigations, PTB treatment did not lead to the restoration of the protein profiles present before *Mtb* infection. Instead, protein profiles of treated patients possess mixed features of PTB and TBL patients. This could indicate that after anti-TB treatment, a pre-existing drug-tolerant *Mtb* population takes over. Evidence from additional studies supports this hypothesis [68].

In addition to the host adaptions, this study also identified proteins of *Mtb* origin in the exosome components, such as DNA-directed RNA polymerase subunit beta RpoC (Rv0668), putative diacyglycerol O-acyltransferase (Rv2285), and possible formate hydrogenlyase subunit (HycE) in circulating exosomes. However, Rv2285 and HycE were also quantified in control patients without active TB. The approach of TMT-based quantification applied here suffers from co-isolation of peptides in MS/MS acquisition [69]. Thus, a co-isolation of peptides of human origin in the spectra of *Mtb* peptides and a false positive reporting of quantitative values in healthy patients cannot be excluded. Of note, the identification of the proteins described here is in line with previous reports, and are related to *Mtb* replicating systems in the host [70], transmission of multidrug-resistant tuberculosis [71], and fatty acid supply [47]. Of note, we observed possible upregulation of formate hydrogenlyase subunit (HycE) in TBL patients. However, to the best of our knowledge, no role of formate hydrogenlyases (FHL) complexes or formate metabolism in the physiology and virulence of *M. tuberculosis* has been previously described and should be explored in future studies.

## 5. Conclusions

This is the first quantitative investigation of exosomes from TB patients’ plasma using lipidomics and proteomics techniques, which have been combined here to provide a better understanding of the host–pathogen relationship. These findings provide a fresh view that *Mtb* infection requires lipids (SM, PC, CE, and TAG) to increase its proliferation in the host and downregulates antibacterial proteins (CAMP, FCN3, SSC5D) as a putative immune evading mechanism. Disease state-specific molecular profiles of circulating exosomes include SMs and TAGs at lipid level, as well as proteins involved and cellular adhesion and the innate immune system. These findings add to the knowledge about the general mechanisms of *Mtb* pathogenicity and putative escape mechanisms to tissues beyond the lungs. Our results show that treatment of PTB patients influences the fatty acyl composition of SM, PC, PI, CE, and TAG, and restore it towards the composition of non-infected volunteers. These findings could not be recapitulated on the protein level, where anti-TB treatment had only minor impacts. However, this analysis may indicate proteins involved in drug-resistance of *Mtb*. Furthermore, this study directly identified *Mtb*-originating protein, which may be utilized for TB diagnostics or vaccination.

This study recapitulates several findings of previous reports. Of note, while previous studies applied invasive sampling techniques or mechanistically in vitro studies, we could confirm this by analyses of circulating exosomes in minimally invasive sampling in a clinical setting. Additionally, this study shows that several proteins are also affected by *Mtb* infection that have not previously been the focus of TB research, including adhesion proteins (e.g., CD151, PRG4), transporters (e.g., ORM1, CLU), and acute phase proteins (SAA1, SERPINA1).

At this point, it should be noted that due to the limitations in terms of patient numbers as well as additional clinical parameters, this study has a pilot character. Further research, including controlled and larger patient cohorts as well as documentation of clinical parameters of patients, is needed in order to increase the significance and confidence of the observed molecular profiles and uncover their functional relevance.

## Figures and Tables

**Figure 1 biomedicines-10-00783-f001:**
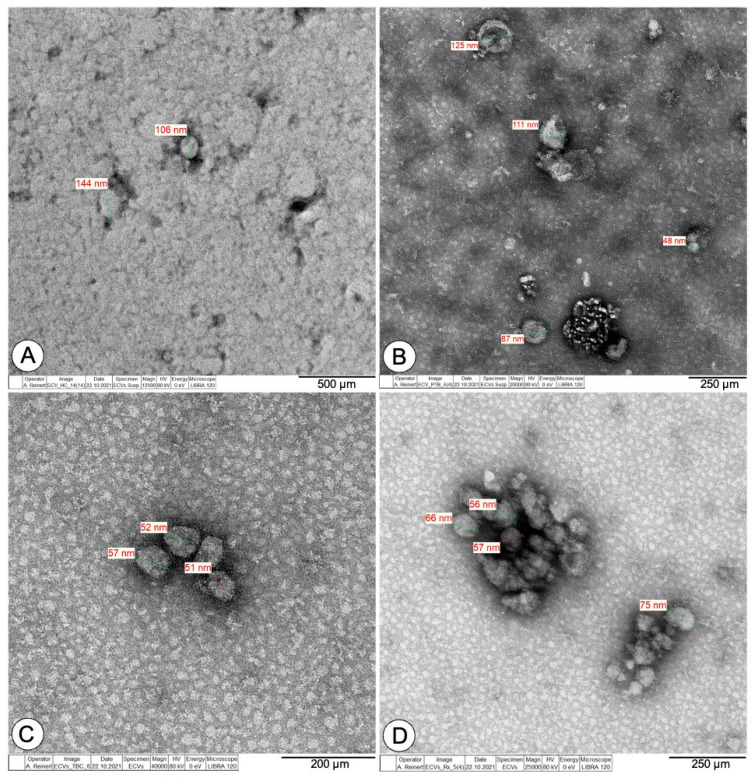
Transmission electron microscopy images of exosome isolates from blood plasma of healthy individuals (**A**), pulmonary tuberculosis (**B**), tuberculous lymphadenitis (**C**), and after anti-TB drug treatment (**D**). One representative picture out of at least three independent experiments is shown for each group.

**Figure 2 biomedicines-10-00783-f002:**
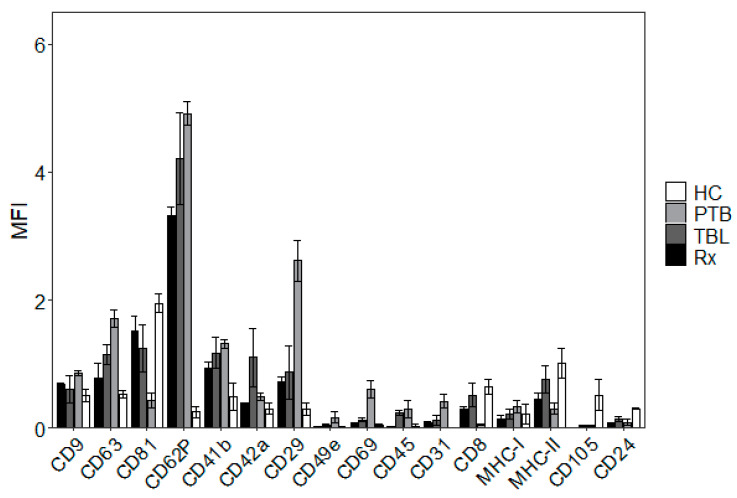
Flow cytometric MACSPlex exosome kit analysis of exosome surface protein markers. The x-axis represents selected proteins, whereas the y-axis shows the normalized APC-MFI. The median APC-signal intensity of each specific population of single beads was normalized to the average of the anti-CD9, anti-CD63, and anti-CD81 beads. MFI, median fluorescence intensity. Data sets are presented as mean ± SEM (*n* = 3). Complete data of all analyzed surface markers is provided in Appendix A.

**Figure 3 biomedicines-10-00783-f003:**
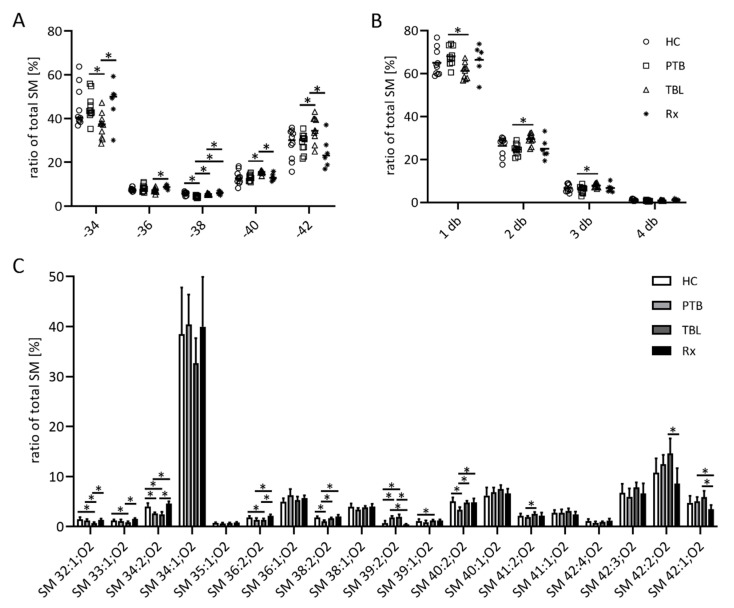
Composition of the sphingomyelin (SM) fraction of exosomes from TB patients. Organic extracts of exosome-derived lipids isolated from patients with pulmonary tuberculosis (PTB) and tuberculous lymphadenitis (TBL) as well as PTB-treated patients (Rx) were separated by high-performance thin layer chromatography. SM-containing spots were automatically eluted with methanol and directly analyzed by ESI-IT MS. The sum of SM species with up to 34, 36, 38, 40 and 42 C atoms (**A**) and the sum of SM species with one, two, three and four double bonds (db) (**B**) as well as the relative amounts of single SM species (**C**) were calculated from the sum of the signal intensities of all SM. Data sets in panels (**A**,**B**) are depicted as dot plots to show all single values, whereas data sets in panel (**C**) are depicted as bar graphs showing the mean and the positive standard deviation. Statistical significance was determined using the unpaired *t* test and the Holm–Sidak method to correct for multiple comparisons (alpha = 0.05). * *p* < 0.05.

**Figure 4 biomedicines-10-00783-f004:**
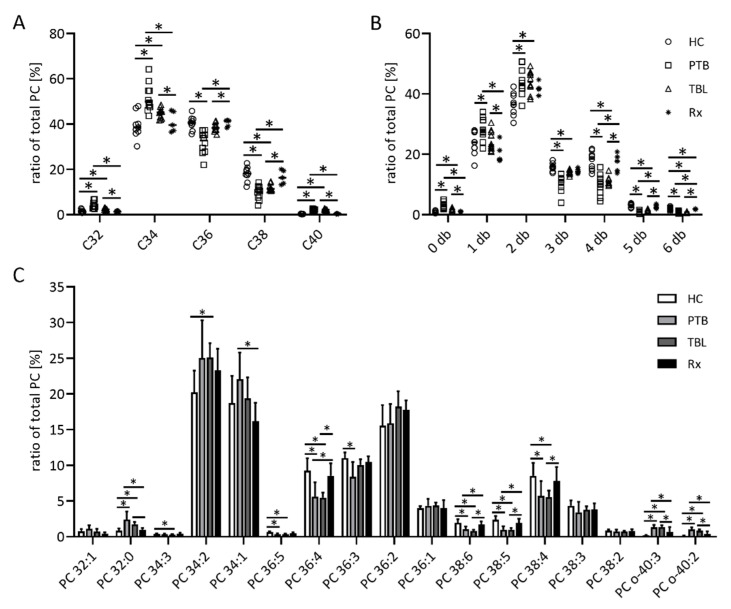
Composition of the phosphatidylcholine (PC) fraction of exosomes from TB patients. Organic extracts of exosome-derived lipids isolated from patients with pulmonary tuberculosis (PTB) and tuberculous lymphadenitis (TBL) as well as PTB-treated patients (Rx) were separated by high-performance thin layer chromatography. PC-containing spots were automatically eluted with methanol and directly analyzed by ESI-IT MS. (**A**) The sum of PC species with 32, 34, 36, 38 and 40 C atoms (**A**) and the sum of PC species with a certain amount of double bonds (db) in their fatty acyl residues (**B**) as well as the relative amounts of single PC species (**C**) were calculated from the sum of the signal intensities of all PC. Data sets in (**A**,**B**) are depicted as dot plots to show all single values, data sets in (**C**) are depicted as bar graphs showing the mean and the positive standard deviation. Statistical significance was determined using the unpaired *t* test and the Holm–Sidak method to correct for multiple comparisons (alpha = 0.05). * *p* < 0.05.

**Figure 5 biomedicines-10-00783-f005:**
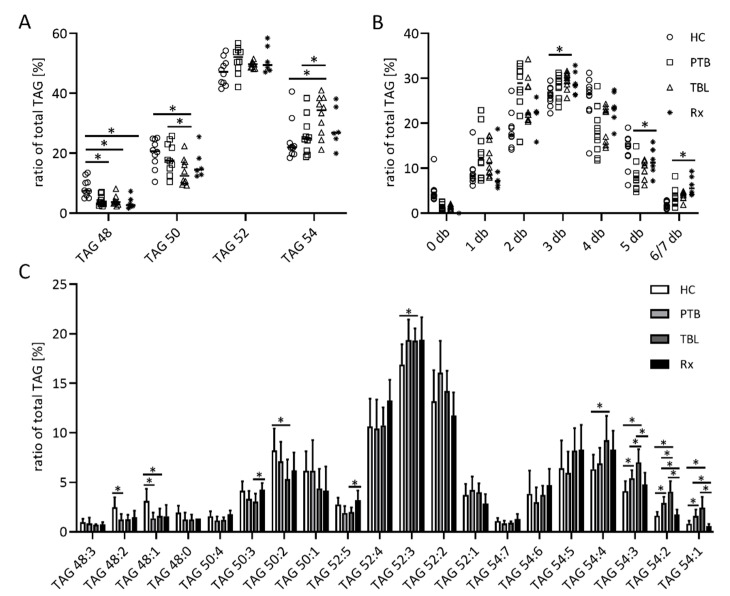
Composition of the triacylglycerol (TAG) fraction of exosomes from TB patients. Organic extracts of exosome-derived lipids isolated from patients with pulmonary tuberculosis (PTB) and tuberculous lymphadenitis (TBL) as well as PTB-treated patients (Rx) were separated by high-performance thin layer chromatography. TAG-containing spots were automatically eluted with methanol and directly analyzed by ESI-IT MS. The sum of TAG species with 48, 50, 52, and 54 C atoms (**A**), the sum of TAG species with a certain amount of double bonds (db) in their fatty acyl residues, (**B**) as well as the relative amounts of single TAG species (**C**), were calculated from the sum of the signal intensities of all TAG. Data sets in (**A**,**B**) are depicted as dot plots to show all single values, data sets in (**C**) are depicted as bar graphs showing the mean and the positive standard deviation. Statistical significance was determined using the unpaired *t* test and the Holm–Sidak method to correct for multiple comparisons (alpha = 0.05). * *p* < 0.05.

**Figure 6 biomedicines-10-00783-f006:**
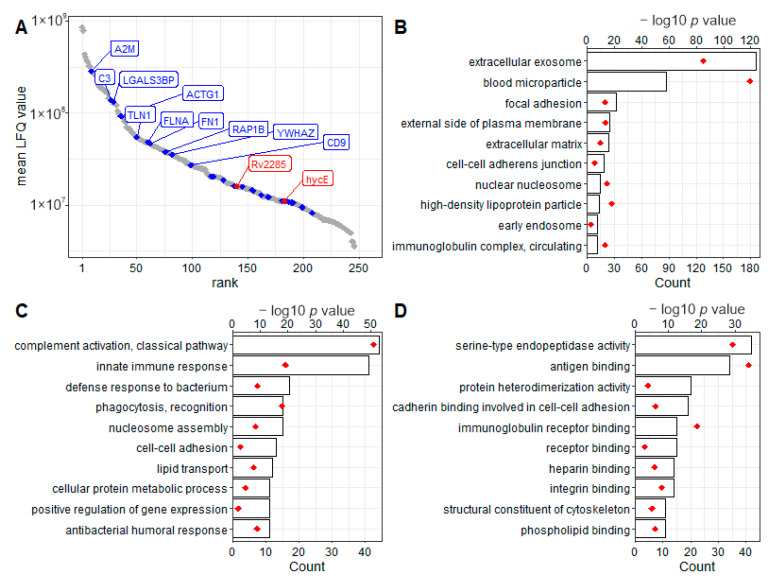
Identification of exosomal proteins by mass spectrometry-based proteomics. (**A**) Proteins were isolated from circulating exosomes, proteolytically cleaved, and analyzed via mass spectrometry-based proteomics. Proteins were identified by MaxQuant (version 1.6.3.3). Protein abundances were estimated by the included MaxLFQ algorithm. Based on label-free quantification (LFQ), isolated proteins span about five orders of magnitude of abundance and include common populations of frequently observed vesicular proteins (blue) and proteins of *M. tuberculosis* origin (red). Identified proteins were used for gene ontology (GO) overrepresentation analysis, including the categories “cellular components” (**B**), “biological processes” (**C**), and “molecular functions” (**D**). Representative terms are illustrated by the number of included proteins (bars) and the significance of overrepresentation (as-log_10_ *p* value, red diamonds).

**Figure 7 biomedicines-10-00783-f007:**
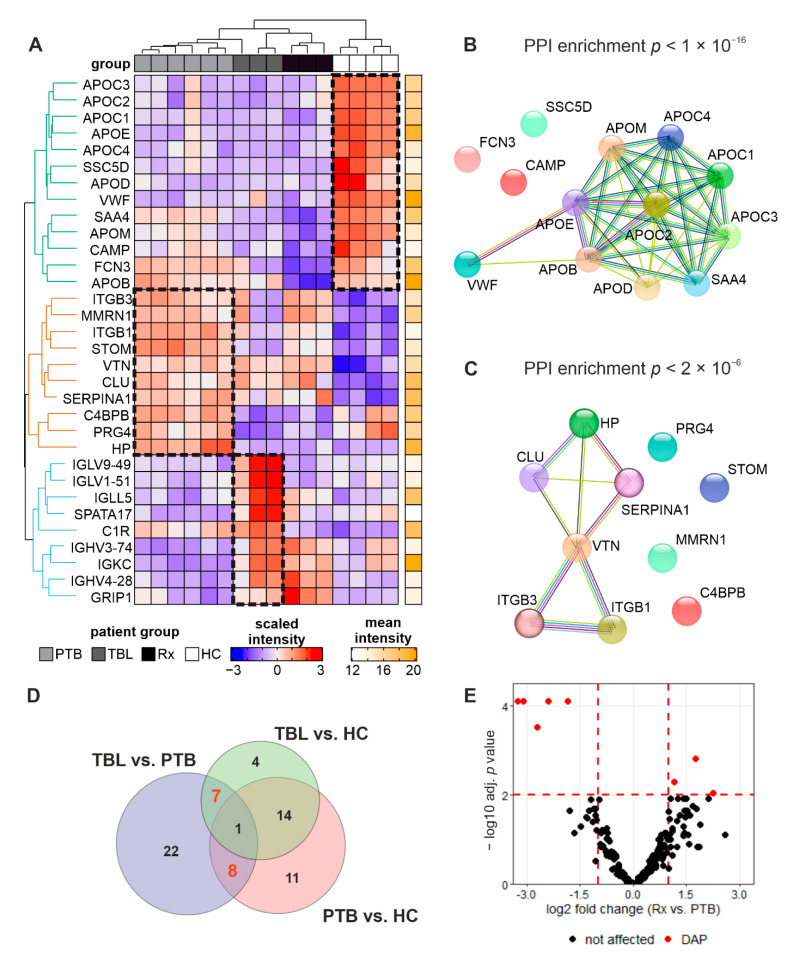
Disease and treatment state-specific protein composition. Comparative analyses of protein abundances at different disease and treatment stages were performed by tandem mass tag (TMT)-based quantification. Hierarchical clustering of ANOVA-revealed differentially abundant proteins (adjusted *p* < 0.01) resulted in a nearly perfect separation of the disease groups (columns) and grouping of proteins into three main branches (rows, branch 1 = green, branch 2 = brown, branch 3 = blue). Protein abundance values in heatmaps were scaled row-wise for visualization (**A**). Functional network analysis with STRING revealed significant protein–protein interaction (PPI) enrichments for proteins of main branch 1 (**B**) and main branch 2 (**C**). Stepwise pairwise analyses revealed 14 common, as well as eight and seven disease state-depending DAP as depicted in a Venn–Euler diagram (**D**). The Volcano plot represents treatment-affected DAP with log_2_-transformed ratios of protein abundances (Rx vs. PTB) and log_10_-transformed Benjamini–Hochberg-adjusted *p* values (**E**). HC—healthy controls, PTB—pulmonary tuberculosis, TBL—tuberculous lymphadenitis, and Rx—PTB after anti-TB drug treatment.

**Table 1 biomedicines-10-00783-t001:** Scheme of tandem mass tag-labeled (TMT) batches. Eighteen samples were labeled with tandem mass tags 10-plex kit (columns, 126–131) and combined into two batches (rows), both including a common reference (com. ref.) for inter-batch correction. HC—healthy control, PTB—pulmonary tuberculosis, TBL—tuberculous lymphadenitis, Rx—PTB after anti-tuberculosis treatment. Numbers behind sample group annotations indicate replicate identity.

TMT-Label	126	127N	127C	128N	128C	129N	129C	130N	130C	131
Batch 1	com. ref.	HC_1	PTB_1	HC_2	PTB_2	HC_3	PTB_3	TBL_1	TBL_2	TBL_3
Batch 2	com. ref.	RX_1	HC_4	RX_2	HC_5	RX_3	HC_6	PTB_4	PTB_5	PTB_6

**Table 2 biomedicines-10-00783-t002:** Summary of disease state and treatment-specific protein contents of circulating exosomes. Proteins identified as differentially abundant proteins (DAP) in ANOVA and pairwise analyses were grouped by specificity to disease state or anti-TB treatment effect. Functional grouping was implemented according to manual literature research.

Depleted in *Mtb* Infection	Increased in PTB	Increased in TBL	Treatment Affected
antibacterial proteins (CAMP, FCN3, SSC5D) apolipoproteins (APOB, APOC1, APOC2, APOC3, APOC4, APOD, APOE, APOM) others (SAA4, VWF)	adhesion proteins (ITGB1, ITGB3, CD151, ICAM2, VTN, MMRN1, PRG4) transporters (ORM1, SLC2A3, HP, CLU) others (SAA1, SERPINA1, C4BPB, STOM)	immunoglobulins (IGHV3-74, IGHV4-28, IGKC, IGLL5, IGLV1-51, IGLV9-49) others (C1R, GRIP1, SPATA17, C9orf50)	increased (IGHV4-28, GRIP1, MYH9) decreased (FCN3, HP, APOB, SAA4, TMEM215)

## Data Availability

The proteomics MS data are available via the PRIDE data repository with the identifiers PXD030883 and 10.6019/PXD030883. Further data is contained within the article.

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
