# Peer review of "Mycobacterium tuberculosis Affects Protein and Lipid Content of Circulating Exosomes in Infected Patients Depending on Tuberculosis Disease State"

_biomedicines, 2022, doi:10.3390/biomedicines10040783_

Round 1

Reviewer 1 Report

This manuscript requires major revision and language editing for clarity and brevity. 

Requires controls for potential contamination of plasma derived proteins and lipids in exosome preparations. 

Discussions and other parts of the manuscript could be shortened with a focus. 

Abstract is long and it is hard to find the main findings of this study.

This manuscript describes a study to profile the proteins and lipids of the exosomes isolated from the plasma samples of healthy subjects, pulmonary tuberculosis patients and the patients with tuberculous lymphadenitis as extrapulmonary tuberculosis to determine the differences in molecular contents of the exosomes with a focus on proteins and lipids.  Although the research topic is interesting and potentially provide very important reference material for the future studies in similar areas of tuberculosis research, the manuscript was not written well. It is hard to follow and some of the descriptions are hard to understand which require language editing and shortening of the manuscript. Besides, there are plenty of literature as reviewed and summarized by the same authors previously, there is not clear description of the novelty of this research in the context of the current literature in the exosome and tuberculosis field. There is no control for the potential contamination of plasma derived proteins and lipids in the exosomes obtained from the study subjects.

Author Response

Please see the file attached.

Reviewer 2 Report

The manuscript by Fantahun Biadglegne et al. presented to me for review, entitled "Mycobacterium tuberculosis affects protein and lipid content of circulating exosomes in infected patients depending on the tuberculosis disease state", deals with the still crucial world topic of tuberculosis, the characteristics of this disease and its diagnosis.

This manuscript has been prepared with appropriate care, and the presented methods and results indicate a well-thought-out design of the experiments. However, despite this care, it has some drawbacks and requires some minor corrections.

  • In line 60, when listing particular types of tuberculosis, the notation musculoskeletal TB was missing in brackets
  • In the materials and methods 2.1, it was indicated that the patients were diagnosed according to the Ethiopian Ministry of Health guidelines. Due to the very different approach to the diagnosis of tuberculosis in the world, I suggest, at least briefly, indicate which tests and procedures are the basis of the diagnosis.
  • Have healthy volunteers been tested for the possibility of latent infection?
  • Line 294, no spaces between "alpha = 0.05.Graphs"
  • The description of figure 2 refers to the wrong table in the supplementary materials.
  • Please check the numbering of the tables in the text and the supplementary materials carefully.
  • Figures and tables in the supplementary materials do not contain appropriate descriptions.
  • Line 393 no punctuation or spaces.
  • Line 577 is an inconsistent abbreviation of MHC class I molecules.
  • Please pay attention to the quality and resolution of the figures included in the manuscript.
  • It seems that a good solution where it is possible will be to replace bar charts with dot plot charts showing the distribution of individual measurements.

Author Response

Please see the file attached.

Round 2

Reviewer 1 Report

Thanks for the revision of your manuscript accordingly.